# Fecal Microbiota Transplantation and Other Gut Microbiota Manipulation Strategies

**DOI:** 10.3390/microorganisms10122424

**Published:** 2022-12-07

**Authors:** Gianluca Quaranta, Alessandra Guarnaccia, Giovanni Fancello, Chiara Agrillo, Federica Iannarelli, Maurizio Sanguinetti, Luca Masucci

**Affiliations:** 1Department of Laboratory and Infectious Sciences, A. Gemelli University Hospital IRCCS, 00168 Rome, Italy; 2Department of Basic Biotechnological Sciences, Intensivological and Perioperative Clinics, Catholic University of Sacred Heart, 00168 Rome, Italy

**Keywords:** gut microbiota modulation, eubiosis, diet, FMT, phages, engineered bacteria, probiotics, prebiotics, personalized medicine

## Abstract

The gut microbiota is composed of bacteria, archaea, phages, and protozoa. It is now well known that their mutual interactions and metabolism influence host organism pathophysiology. Over the years, there has been growing interest in the composition of the gut microbiota and intervention strategies in order to modulate it. Characterizing the gut microbial populations represents the first step to clarifying the impact on the health/illness equilibrium, and then developing potential tools suited for each clinical disorder. In this review, we discuss the current gut microbiota manipulation strategies available and their clinical applications in personalized medicine. Among them, FMT represents the most widely explored therapeutic tools as recent guidelines and standardization protocols, not only for intestinal disorders. On the other hand, the use of prebiotics and probiotics has evidence of encouraging findings on their safety, patient compliance, and inter-individual effectiveness. In recent years, avant-garde approaches have emerged, including engineered bacterial strains, phage therapy, and genome editing (CRISPR-Cas9), which require further investigation through clinical trials.

## 1. Introduction

### 1.1. Gut Microbiota Composition

Trillions of different microorganisms, bacteria, archaea, phages, and protozoa compose the gut microbiota. It represents a solid organ, weighing approximately 2 kg, and able to direct host metabolic and immunity pathways [1]. Specifically, the gastrointestinal tract harbors approximately 10^13^–10^14^ bacterial cells [2]. The most represented phyla are Firmicutes, Bacteroidetes, and Proteobacteria, while Actinobacteria contribute less to the total bacterial composition [1,3]. The Firmicutes phylum is composed of more than 200 genera, of which Clostridioides is the most represented (95%). On the other hand, in the Bacteroidetes phylum, *Prevotella* and *Bacteroides,* are the predominant genera, while the Bifidobacterium genus represents the major member in the Actinobacteria phylum [4]. Bacterial species are variably distributed throughout the gastrointestinal tract. *Streptococcus, Prevotella, Actinomyces, Gemella, Rothia, Granulicatella, Haemophilus*, and *Veillonella* are the predominant genera in the throat and distal esophagus [5]. In the stomach, microbial diversity depends upon the presence and absence of Helicobacter pylori [6,7]. Stomachs lacking *H. pylori* are mainly populated by *Streptococcus* spp., *Actinomyces* spp., *Prevotella* spp., and *Gemella* spp., which are predominantly found in the throat, indicating that they may be transient residents originating there [5]. The recto-sigmoid colon microbiota is more complex than the jejunum, ileum, and caecum resident microbes. *Enterococcus* spp., *Escherichia coli*, *Klebsiella* spp, *Lactobacillus* spp., *Staphylococcus* spp., and *Streptococcus* spp. are present in the jejunum and ileum. Most microbes of the jejunum and ileum are aerobes and facultative anaerobes [8]. The small intestine harbors the aerobic *Enterococcus* group, *Lactobacillus* spp., *Streptococcus* spp., and Gammaproteobacteria, whereas anaerobes are predominant in the large intestine. Cecal microbiota are more complex than jejunal and ileal microbiota. The cecal bacteria are predominated by *Lactobacillus*, *Enterococcus*, and *E. coli* [9]. In the recto-sigmoidal colon, strictly anaerobic bacteria belonging to *Bacteroides* spp., *Clostridioides coccoides*, and *Clostridioides leptum* are the predominant bacterial groups [8]. Given the complexity and multifactorial activities in terms of the evolution of the human intestinal microbiota, it is difficult to establish the composition of an ideal and healthy microbiota. Generally, a state of eubiosis is characterized by a strong presence of Firmicutes and Bacteroidetes and by a low percentage of Proteobacteria which, instead, increase during inflammatory states.

### 1.2. The Gut Microbiota and Health/Disease Status

In the last decade, considerable interest has focused on the gut microbiota in certain host physiopathological balances [10]. Nowadays, the assessment of gut bacterial composition is the primary step for investigating potential links between gut microbes and hosts. Several studies have demonstrated bacterial unbalance associated with various intestinal and systemic diseases such as obesity, inflammatory bowel disease (IBD), and cardiovascular, neurological, and psychiatric disorders, highlighting a key role of gut microbes in the physiopathogenesis of intestinal and extraintestinal diseases [11]. Due to the main investigative methodologies, metagenomics, and culturomics, researchers have been able to clarify the potential pathogenic role of specific bacterial species, found in samples of healthy and afflicted patients. For example, butyrate-producing bacteria such as *Faecalibacterium prausnitzii*, *Akkermansia muciniphila*, *Roseburia*, and *Eubacterium*, are considered to be beneficial species that lead to improvements in gut barrier integrity and compete with pathogenic bacteria [12]. On the other hand, specific microbial profiles are associated with certain clinical conditions. Starting from the well-known linkage between *Fusobacterium nucleatum* and colorectal cancer (CRC), other microorganism–disease associations have been evaluated [13]. For instance, *Streptococcus* spp., *Klebsiella* spp., *Collinsella aerofaciens*, and *Proteus vulgaris* show high abundance in patients with atherosclerosis [14]. Moreover, rheumatoid arthritis patients harbor high levels of *Prevotella copri*, which seems to be the main trigger factor of an altered immune response [15].

In the “personalized medicine era”, understanding the potential impact of bacteria–host relationships is the main aim for researchers. Once identified, the main challenge is to intervene in the modulation of intestinal microbiota in order to modify its composition and, consequently, the functional outputs. In this review, we present an overview of the main present and future strategies influencing the gut microbiota ecosystem, and report data from some clinical trials to describe the state of the art. This review highlights different therapeutical options by which microbial populations may be altered to achieve beneficial effects.

## 2. Microbiota-Changing Strategies

The human gut microbiota is a dynamic entity that has evolved with hosts, and it changes constantly during our lifetime. Initial bacterial imprinting depends on the type of delivery [16]. In fact, in cases of vaginal delivery, early colonizers originate from the mother’s vaginal and fecal microbiota such as *Lactobacillus*, *Prevotella*, *Bacteroides*, and *Escherichia*/*Shigella* [12,16]. In the case of a Caesarean section (C-section), early colonizers belong to the surrounding environment and the mother’s skin microbiota, such as *Staphylococcus* spp., *Streptococcus* spp., *Corynebacterium* spp., and *Propionbacterium* spp. [17]. Further modifications in terms of composition occur during subsequent stages of life, depending on environmental, lifestyle, and nutritional factors, as well as host characteristics (Figure 1). At the age of three years, the composition of the gut microbiota reaches maturity and stability, which is preserved until adulthood. In healthy conditions, this stability is represented by an equilibrium between five different Phyla: Firmicutes (79,4%), Bacteroidetes (16.9%), Actinobacteria (2.5%), Proteobacteria (1%), and Verrucomicrobia (0.1%). When these bacterial communities cohabit peacefully with the host, providing health benefits, “eubiosis” status is reached [18,19].

These factors highlight the intrinsic dynamism of the gut microbiota, expressed through a gradual process [12]. The synergistic action of these elements leads to a metabolic state of health or disease [20]. In recent years, many efforts have focused on supporting this bacterial elasticity, with the aim of intervening in the composition of the intestinal microbiota and, therefore, on the state of health of the host. This changing intervention can be classified into two methodological strategies [20]:Non-specific interventionsSpecific interventions

Non-specific interventions impact the general composition and functions of the intestinal microbiota, enabling the re-establishment of eubiosis conditions. Crucial elements of this strategy are diet, the administration of prebiotics, probiotics, and fecal microbiota transplantation (FMT) [3,17,20]. However, all these strategies could also be directed to specific elements, such as bacterial species and metabolic patterns, to determine select modifications in the function framework [20]. The scenarios described in the literature involve the use of engineered bacteria able to produce metabolic precursors which impact specific metabolites or enhance the competition against pathogens. Moreover, some trials have focused on the use of bacteria for drug delivery and phage therapy in order to delete specific “bad” bacteria populations [21].

### 2.1. Diet

Geographical localization, culture, and genetic background represent crucial elements affecting the microbiota composition and modulation [17,22]. Nevertheless, in recent years, several studies have shown that diet is the main factor responsible for more than 50% of the variability in the microbiota, and that host genetics have a minor role in determining the microbiome composition [23]. Several population-based studies have reinforced this evidence. For example, Westernized and thus most urbanized populations are characterized by reduced microbial biodiversity, with a decline in particular genera, such as *Treponema* [3]. Conversely, non-Westernized rural populations, such as the Hazda community in Tanzania or the Hutterites in North America, whose diets derive from raw or wild foods, exhibit a high microbial diversity and abundance of certain genera, such as *Prevotella* [3,24]. Exploring the link between diet and gut microbial populations is fundamental to outlining the potential of dietary interventions to modulate the composition of the gut microbiota. In general, it is possible to assert that each macronutrient has a specific impact on the gut microbiota, affecting bacterial classes with particular metabolic functions [12,22]. Specifically, a high-fiber diet leads to a marked increase in species that produce short-chain fatty acids (SCFAs), such as *Faecalibacterium prausnitzii*, *Roseburia*, *Eubacterium*, and *Blautia*, which are involved in the fermentation of these fibers in the distal part of the colon. These species are beneficial for eubiosis, contributing to metabolic benefits [25]. Firstly, SCFAs are an important energy source for colonocytes and balancing factors in the intestinal microenvironment by regulating mucus production and lowering the pH in the lumen, resulting in an intact gut barrier and the inhibition of colonization. Furthermore, SCFAs interact with cellular receptors, such as GPCR-41/43 (G-protein-coupled receptors) expressed on L-cells, stimulating the production of PYY (peptide YY) and GLP-1 (glucagon-like peptide 1) hormones, resulting in increased insulin secretion and decreased blood sugar [3,26]. Butyrate is responsible for maintaining an anaerobic environment through the consumption of oxygen mediated by the PPARγ receptor (peroxisome proliferator-activated receptor-γ) [27]. Moreover, butyrate functions in the regulation of cellular proliferation, differentiation, and apoptosis. In particular, this metabolite shows contradictory anti-proliferative effects on differentiated and undifferentiated colonocytes, a phenomenon called the “Butyrate paradox”. In fact, if, in epithelial cells, the anti-proliferative signal represents a protective effect for colonic neoplasia, it has the opposite significance for intestinal stem cells, where anti-proliferative signals are involved in the crypt cells [18,28]. In contrast, a high-protein, high-fat, low-fiber diet, typical of urbanized populations, is associated with reduced biodiversity and an increase in potentially inflammatory species, such as *Bacteroides* [17,29,30]. Specifically, the microbiota of individuals consuming this type of diet is largely characterized by Gram-negative bacteria, and thus strongly induce inflammation via lipopolysaccharide (LPS). The consequent reduction in SCFAs leads to a drop in hormone production and, thus, to a leaky mucosa. Moreover, in this environment, gut bacteria produce trimethylamine (TMA) by metabolizing L-carnitine and phosphatidylcholine [31]. These molecules are ingested from red meat or energy drinks. TMA is then converted in the liver into trimethylamine *N*-oxide, which contributes to atherosclerosis, altering cholesterol metabolism [32,33].

Another crucial factor implicated in the relationship between the diet of the host and gut microbiota composition is represented by bile acid metabolism. Bile acids are molecules synthesized in the liver from cholesterol to then be released in the small intestine by the biliary tract to enhance the absorption of fat-rich molecules and vitamins. While 95% of bile acids are reabsorbed by enterocytes in the ileum and transported back to the liver, the remaining 5% is converted to secondary bile acid by colonic microbial populations, and subsequently reabsorbed into the portal circulation. In this cycle, known as “hepatic circulation”, bacteria and biliary acids play critical roles in hepatic synthesis, creating a negative feedback system. Specifically, intraluminal biliary acids interact with the farnesoid X receptor (FXR), inducing fibroblast growth factor 19 (FGF19), which inhibits biliary acid synthesis in hepatocytes. Additionally, FXR is involved in the secretion of antimicrobial peptides (AMPs) which are antimicrobial agents contrasting bacterial overgrowth. Gut bacteria metabolize primary biliary acids in secondary biliary acids, which have a different affinity with FXR, causing depotentiation to the antibacterial effect [34]. Another role of gut bacteria in human metabolism is their ability to regulate the levels of choline. Choline is a component of very-low-density lipoprotein (VLDL); therefore, its decrease results in a risk of fatty liver due to a triglyceride build-up [35]. Additionally, the product of choline metabolism in trimethylamine-*N*-oxide (TMAO) affects the synthesis of biliary acids. Therefore, a low intake of this metabolite or its consumption by gut bacteria can result in fatty liver conditions [18].

Thus, considering an alimentary regimen only focused on weight loss is a restricted concept. Diet must be considered a key factor in host homeostasis not only in the gut, but also at the systemic level (Figure 2).

These three-way interactions between the host, diet, and microbiota are very complex and remain partly unclear. A short-term diet change can induce a shift in the gut population; however, these changes appear to be transient [17]. Conversely, a prolonged and scheduled diet can induce more stable alterations in the gut microbiota [22,36,37]. This evidence remains unclear, mainly due to a lack of long-term human dietary interventions, or long-term follow-ups of short-term dietary interventions [38]. Discovering biomarkers linked to health or disease is an intriguing challenge for a personalized dietary combination. Obviously, when setting up a targeted diet, it is necessary to determine the clinical parameters, genetic factors, and composition of the microbiota in order to achieve beneficial results in terms of the treatment of certain disorders, such as controlling blood sugar concentrations in diabetic patients, in the prophylaxis of alterations, such as atherosclerosis and cancer treatment [3,22,38]. Ultimately, further investigations to identify factors clarifying the interactions between nutrition and microbiota are required.

### 2.2. Probiotics, Prebiotics, Synbiotics, and Postbiotics

The history of probiotics is as old as human history, and they are currently one of the most used tools to intervene in the gut microbiota. At the beginning of the twentieth century, the Russian biologist E. Metchnikoff, studying a Bulgarian population, noted that the constant intake of yogurt led to an increase in life expectancy [39]. At the time, it was a novel discovery; today, it is well known that fermented foods such as kefir, miso, and sourdough bread are rich in fermenting microorganisms which, when ingested alive, perform various beneficial functions in the human intestine [40]. The current definition establishes that a probiotic is a bacterial strain(s) that has a beneficial effect on the health of the individual who uses it [41]. These mainly include *Bifidobacterium* and *Lactobacillus* strains, as well as some *Enterococcus* and *Streptococcus* strains [42]. These bacterial strains are usually administered with a non-targeted approach in order to improve the balance of the host’s intestinal microbiota. Contemporary evidence has shown that probiotics may induce beneficial and selective effects in the host organism, acting at a systemic level as well. In this regard, Kalliomäki et al. demonstrated that perinatal administration of the probiotic *Lactobacillus rhamnosus* strain GG reduced the incidence of atopic eczema in at-risk children during infancy [43]. Probiotics are also useful for the enhancement of the immune response, and they may have anti-inflammatory effects. Moreover, *Lactobacillus rhamnosus* GG can prevent cytokine-induced apoptosis in colon cells through activation of the antiapoptotic *Akt* oncoprotein and protein kinase B, and inactivation of the proapoptotic *p38* mitogen-activated protein kinase signaling pathway [44]. Furthermore, probiotics play an important role in mental health disorders. Studies in rodent models of depressive disorders have shown that some probiotics inhibit inflammation, decreasing corticosterone levels. In particular, several *Lactobacillus* spp., such as *Lactobacillus plantarum*, *Lactobacillus paracasei*, *Lactobacillus brevis* [45], and *Lactobacillus rhamnosus* [46], change the levels of neurometabolites, such as GABA, leading to mitigation of the symptoms of depressive status. The same action seems to be due to *Bifidobacterium* isolated in humans [47]. To date, a particular focus must be given to a new generation of probiotics (NGP). These especially concern bacterial strains whose effects on the host have been studied in specific pathologies to define their targeted effect on defined pathways [48]. One of the most studied strains is *Faecalibacterium prausnitzii*, a species related to healthy microbiota. These are Firmicutes bacteria and represent about 5% of the healthy gut microbiota [49]. Otherwise, *F. prausnitzii* is almost absent in patients with Crohn’s disease, irritable bowel syndrome (IBS), and coeliac disease, and it has become an important tool to counteract symptoms of these pathologies [49]. Sokol et al. conducted in vivo and in vitro studies to observe the anti-inflammatory activity of *F. prausnitzii* in Crohn’s disease patients; they found that it can induce relatively high amounts of IL-10 and low amounts of IL-12 in peripheral blood mononuclear cells [50]. It is capable of fermenting glucose and producing a variety of SCFAs [51]. Thus, hypothesizing the use of *F. prausnitzii* as a probiotic could promote the production of butyrate, formic acid, and D-lactate, which are useful for maintaining a healthy microbiota [51,52].

Another bacterium whose probiotic activities are now evident is *Akkermansia muciniphila*. In the microbiota of obese and diabetic individuals, it is reduced compared with healthy individuals [53,54]. Everard et al. used a model of obese mice and mice with type 2 diabetes, feeding them a high-fat diet, in which the restoration of mucus on the intestinal mucosa was observed after the administration of *Akkermansia muciniphila* as a probiotic [55,56].

Probiotics are now used with oral somministration in the context of various pathological conditions, but there are some questions about their safety and efficacy. One of the queries is whether the orally assumed bacteria can reach the gut and remain there permanently. Different studies have shown that the concentrations of some strains decrease over time after intake, which implies the need for repeated administration [18,57]. The concept of “colonization resistance” of the gut microbiota to exogenous micro-organism is well known [58].

Another question about the use of probiotics is the possibility of adverse effects related to the introduction of probiotic bacteria, or contaminants, potentially producing toxins [59]. However, these adverse events have been described in a few reports, among which included a case of sepsis by *Lactobacillus* [59] and death from gastrointestinal mucormycosis (derived from contamination) [60].

In the same way, prebiotics have been widely used over the years. They are defined as a naturally occurring ingredient that, when selectively metabolized, induces specific changes in the composition and activity of the intestinal microbiota [41]. The most promising prebiotics for host health are non-digestible oligosaccharides, fructo-oligosaccharides (FOS), galacto-oligosaccharides (GOS), trans-galacto-oligosaccharides (TOS), lactulosem, and inulin [61,62,63]. These elements have peculiar characteristics, including resistance to the acid pH of the stomach, the ability to resist the lytic action of digestive enzymes, and avoiding adsorption by the intestine. Therefore, they can be processed by the resident microbiota [61]. *Lactobacillus* spp., *Bifidobacterium* spp., *Faecalibacterium prausnitzii*, *Anaerostipes* spp., and *Bilophila* spp. are the species most stressed when using these substrates to produce metabolites, including SCFAs [64].

Consequently, prebiotics may improve benefits not only for the gastrointestinal tract. For example, Nicolucci et al. conducted a single-center, double-blind, placebo-controlled trial of two separate cohorts of children with obesity, on whom they tested the use of oligofructose-enriched inulin as a prebiotic, observing decreases in weight, IL-6 levels, and triglyceride levels in those grouped in the treatment cohort [65]. As described above, SCFAs are key metabolites for intestinal structural and functional homeostasis. A key challenge to be addressed is the clarification of the impact of prebiotics on the levels of SCFAs produced. Holmes et al. (2022) conducted their study to evaluate the response of the gut microbiota to different prebiotics. In this study, the authors administered the prebiotics inulin, GOS, and dextrin to a set of 28 individuals in a three-period cross-over design, consisting of a prebiotic-free week followed by a prebiotic-supplemented week. The data revealed increased SCFA levels at the population level, although inconsistent results were described at the individual level. Indeed, this variation in individual response was associated with differences in microbiota composition at baseline and dietary factors. Specifically, the researchers found that butyrate production in response to the prebiotic was inversely proportional to habitual dietary fiber intake [66]. This confirms that the composition of the microbiota is the main factor in SCFA metabolism. The responses to prebiotics appear to be conserved at the population level; therefore, the efficacy of therapy could be optimized by specifically selecting patients with specific microbial profiles [67].

Great interest was aroused by the synergistic use of probiotics and prebiotics to enhance bacterial vitality and their beneficial capacities. These products are defined as synbiotics [68]. To date, the functionalities and advantages of synbiotics in treating some diseases, such as obesity, are being evaluated. Preliminary in vitro studies have shown a greater efficiency of synbiotics in modulating the microbiota compared with prebiotics and probiotics used separately [69].

Another category of elements capable of acting on the gut microbiota, regulating its functionality, and benefitting the host, is represented by postbiotics. The concept of postbiotics arose from the observation that the beneficial effects of the microbiota on the host mainly depend on the specific metabolites produced by bacteria populations. Therefore, the goal is to have industrially produced metabolites taken up to act directly on the host’s cells, without undergoing the metabolic activity of microorganisms [70]. There are many sources and methods for obtaining postbiotics in vitro. For example, in the supernatants of *Lactobacillus acidophilus* and *Lactobacillus casei*, antioxidant and anti-inflammatory molecules act on the intestinal epithelium cells [71]. Other examples of molecules that can be defined as postbiotics are some types of bacterial enzymes acting on reactive oxygen species, such as glutathione peroxidase and peroxide dismutase produced by *Lactobacillus fermentum E-3* and *E-18*, strains isolated from intestinal populations of a healthy child [72]. Otherwise, *Lactobacillus helveticus* produces phenolic-derived metabolites, aromatic amino acids, vitamins, and SCFA molecules, which are fundamental for the maintenance of intestinal homeostasis and for beneficial effects at the systemic level [70,73].

All this evidence suggests that bacteria and their metabolites may play a key role in human gut homeostasis. Their use as a therapeutic tool involves numerous advantages, but also strong limitations. Probiotics, prebiotics, synbiotics, and postbiotics have several advantages. They are safe drug products, easy to administer, and have no severe adverse events. Conversely, the weaknesses are the transient beneficial effects depending on the host’s resident microbial populations and on “colonization resistance” involving continuous administration over time [58,74].

### 2.3. Fecal Microbiota Transplantation (FMT)

One of the most innovative therapeutic approaches for gut microbiota modulation is represented by FMT. FMT consists of the infusion of fecal suspension obtained from a healthy donor into the bowel of a recipient patient in order to restore microbial population equilibrium [75]. The origins of this method date back to the fourth century. It was an ancient Chinese medicine practitioner named Ge Hong who first used a feces suspension called “yellow soup” to treat patients with severe diarrhea [76]. No further FMT reports were noted until the 17th century when veterinarians used stool as a therapeutic option for livestock farms. Camel stool was also used by German soldiers to treat bacterial dysentery during World War II [77]. The modern era of FMT as a therapy can be traced to 1958. Eiseman and colleagues treated four cases of pseudomembranous colitis with a fecal suspension administered via enema [78]. In the last decade, FMT bounced back dramatically as a valid therapeutic solution. The intestinal microbiota of *Clostridioides difficile* infection (CDI) patients is characterized by a marked increase in Proteobacteria and a strong decrease in the Firmicutes/Bacteroidetes ratio [79]. Thus, the primary objective of fecal transplantation is to reverse this microbial pattern, re-establishing a state of eubiosis. CDI was the first disorder treated by FMT. The first randomized controlled clinical trial was performed in the Netherlands by Van Nood et al. in 2013. In this study, the authors observed marked clinical resolutions of 81% and 94% after one and two FMT procedures, respectively, compared with 31% after the administration of vancomycin [80]. This work represented the first of many clinical trials on CDI. In several subsequent publications, FMT via colonoscopy has been shown to be more efficient than fidaxomicin and vancomycin regimens [81,82,83]. The treatment of CDI by FMT was only the first step for its use as a valid clinical therapy. Considering the well-described gut–brain axis and the complex pathways exerted by microbial populations, FMT represents a potential tool for the treatment of extra-intestinal disorders. Very interesting findings were observed in neurological, cardiovascular, and metabolic disorders [84].

#### 2.3.1. FMT and Cardiovascular Disease

In recent years, gut microbiota transplantation has also become a subject of therapeutic interest in cardiovascular diseases (CVDs). The scientific literature has produced numerous papers demonstrating a close link between gut bacteria and the development and progression of circulatory diseases, such as atherosclerosis and hypertension [85]. Particularly, it is well known that TMAO production in the liver, influenced by the gut microbiota, is a key mechanism of CVD. This metabolite has been demonstrated to modulate the cholesterol and sterol regulation pathways, cholesterol transport, and bile acid levels [85]. Elevated serum levels of TMAO are associated with early atherosclerosis and with a high risk of CVD mortality [33]. An interesting study was conducted by Kim et al. in 2022 [86]. The purpose of their study was to determine the impact of the gut microbiota on the pathogenesis of atherosclerosis caused by genetic deficiency using animal models represented by C1q/TNF-related protein 9-knockout mice (CTRP9-KO) and wild-type (WT) mice. To outline the association between the gut microbial population and atherosclerosis progression, cross-fecal transplantations were performed. WT mice were inoculated with fecal material derived from CTRP9-KO mice and CTRP9-KO mice were inoculated with fecal material from WT mice. This study showed that the transplantation of WT microbiota into CTRP9-KO mice protected against the progression of atherosclerosis. Conversely, the transplantation of CTRP9-KO microbiota into WT mice promoted the progression of atherosclerosis. Moreover, atherosclerotic lesions in the carotid arteries were decreased in transplanted CTRP9-KO mice compared with CTRP9-KO mice prior to transplantation. Conversely, WT mice transplanted with the gut microbiota of CTRP9-KO mice showed the opposite effect [86].

Another relevant CVD is hypertension. It is the most prevalent cardiovascular disease, affecting ~30% of the adult population worldwide [87]. Zhong et al. (2021) conducted a study demonstrating that microbiota transplantation from healthy donors can decrease blood pressure in patients with hypertension [88]. The data showed that microbiota transplantation determined a marked antihypertensive effect on blood pressure compared with normotensive patients. In fact, the blood pressure values at hospital discharge were significantly lower than those at hospital admission (change in systolic blood pressure: −5.09 ± 15.51, *p* = 0.009; change in diastolic blood pressure: −7.74 ± 10.42, *p* < 0.001). Moreover, researchers have investigated better FMT delivery routes, comparing “upper” and “low” gastrointestinal tracts. Data have shown that hypertensive patients subjected to FMT via the lower gastrointestinal tract (β = −8.308, standard error = 3.856, *p* = 0.036) have a greater decrease in systolic blood pressure [88].

#### 2.3.2. FMT and Autism

Several studies have suggested that gut microbiota composition plays a fundamental role in gastrointestinal and neurodevelopmental dysfunction in autism spectrum disorder (ASD) patients [89,90,91]. In 2019, Kang et al. described the benefits of microbiota transplantation therapy on autism symptoms. They performed an open-label trial of microbiota transfer therapy (MTT) which combined antibiotics, a bowel cleanse, a stomach acid suppressant, and fecal microbiota transplant on 18 patients, with a follow-up period of two years. Significant improvements in gastrointestinal symptoms, autism-related symptoms, and gut microbiota composition were observed. Interesting findings indicated that most improvements in gastrointestinal symptoms were maintained and autism-related symptoms improved, even after the end of the treatment. Important changes in gut microbiota composition at the end of the treatment remained at follow-up, including significant increases in bacterial richness and the relative abundance of genera *Bifidobacteria* and *Prevotella* [90]. Further encouraging evidence was described by Li et al. in 2021. They conducted a clinical trial involving 40 ASD and 16 control children, evaluating the effect of gut microbiota infusion and oral capsules on gastrointestinal and ASD symptoms. They found a marked difference in baseline characteristics of behavior, intestinal symptoms, and microbiota composition between ASD and controls. Specifically, symptoms including abdominal pain, indigestion, and diarrhea were significantly improved after FMT treatment had persisted for 8 weeks. Oral and rectal routes showed similar beneficial effects on ASD children. Moreover, parameters such as mood, behavior, language, and social skills were also ameliorated after treatment, although these improvements were reversed after 8 to 12 weeks without further treatment [91].

#### 2.3.3. FMT and Depression

Interactions between gut bacteria and the central nervous system (CNS), through the so-called “gut–brain axis”, are now well known. Several investigations have shown that the synthesis of neurochemical molecules, such as serotonin receptor expression, is closely linked to gut microbiome alterations [92]. In recent years, the challenge has been to improve some psychiatric conditions through the modulation of gut microbial populations. Data from animal models have highlighted the key role played by FMT in reversing neuroinflammation and depression-like behavior and improving anxiety [93]. Given the number of patients worldwide (264 million people) affected by major depressive disorder (MDD) and the associated failed therapies, depression represents a key therapeutic challenge [94]. Doll et al. (2022) demonstrated the improvements achieved after FMT was performed on two patients with MDD. The protocol included the administration of 30 oral frozen FMT capsules within 90 min under the observation of a physician. Each active 30-capsule dose consisted of 8.25 g of donor stool, originating from a different donor for each patient. After 4 and 8 weeks, post-intervention measurements were conducted on symptoms and fecal samples. The evidence showed that both patients exhibited improvements in their depressive symptoms 4 weeks after the transplantation, and beneficial effects lasted for up to 8 weeks in one patient. Moreover, considerable improvements in gastrointestinal symptoms were observed [95].

Particularly, in one patient, decreases in *Victivallis*, *Alistipes*, *Roseburia*, *Prevotella*, *Ruminococcus*, *Blautia*, and *Faecalibacterium*, and the family Lachnospiraceae, were observed after the FMT procedures were performed. In the second patient, decreases were noted in the genera *Ruminococcus*, *Alistipes*, *Bifidobacterium*, and *Oscillibacter*, and the family Lachnospiraceae.

#### 2.3.4. FMT and Sclerosis Multiple

Multiple sclerosis (MS) is an autoimmune, inflammatory, demyelinating disease of the CNS influenced by genetic susceptibility and environmental factors [96]. Several studies have demonstrated that MS patients have numerous taxonomic alterations in their gut microbiota composition, including relative increases in *Pseudomonas*, *Blautia*, *Streptococcus*, and *Akkermansia* spp., and decreases in *Prevotella*, *Bacteroides*, *Parabacteroides*, and *Clostridia* spp., compared with healthy individuals [97]. These findings suggest the role of microbiota as a therapeutic target for MS. The first interesting article focused on FMT; clinical MS improvements date back to 2011. Borody at al. described the cases of three patients with MS diagnoses who achieved durable symptom reversal for constipation with FMT. In all cases, the authors reported the complete resolution of constipation symptoms after five FMT procedures. Very intriguing were the progressive improvements in neurological symptoms, regaining the ability to walk following the slow resolution of leg paresthesia, and increased energy levels [98]. In the following years, several studies have been conducted in order to clarify the real link between gut microbes and MS.

Kait et al. (2022) performed a pilot randomized controlled trial enrolling nine patients. The patients were provided monthly FMT for up to six months in order to outline FMT safety and tolerability in MS and the effectiveness of treatment on abnormal intestinal permeability. Kait et al. demonstrated that FMT was safe in this group of patients, reporting no adverse events. Moreover, two patients with elevated small intestinal permeability at baseline improved to normal values following FMT treatment [99]. The MS patients showed a non-significant change in bacterial alpha diversity following multiple FMTs. Other studies report no difference in microbiota composition between MS patients and healthy controls.

#### 2.3.5. FMT and Metabolic Disease

Metabolic disorder represents one of the most widely investigated fields, in which gut microbes and systemic host health are closely associated. Building on the evidence that bacterial compositions differ between lean and obese animals, and that the gut microbial composition may reflect an aberrant metabolic function [98], many efforts have been directed towards the beneficial modulation of gut populations in order to restore a “homeostatic” state. As described above, several research groups have focused on the potential use of FMT to impact metabolic alterations. Another interesting example is the study conducted by Koote et al. in 2017, in which the effects of the lean donor (allogenic) versus personal (autologous) FMT on 38 male recipients with metabolic syndrome were investigated. Patients were followed for 18 weeks after the interventions. Researchers have evaluated the metabolic parameters and the gut microbial composition in the short and long term. Specifically, long-term changes in the major outcome parameters have not been observed; conversely, between weeks 0 and 6, it was found that allogenic FMT resulted in altered duodenal and fecal microbiota compositions. This was associated with improved peripheral insulin sensitivity at week 6 (from 25.8 [19.3–34.7] to 28.8 [21.4–36.9] μmol kg^−1^ min^−1^, *p* < 0.05), whereas autologous FMT had no effect (from 22.5 [16.9–30.2] to 20.8 [17.6–29.5] μmol kg^−1^ min^−1^). The change in peripheral insulin sensitivity in the allogenic FMT group was accompanied by a small but significant decrease in glycated hemoglobin (HbA1c) at 6 weeks (from 39.5 [36.0–41.0] to 38.0 [34.0–41.0] mmol/mol, *p* < 0.01). Concerning fecal microbial diversity, no significant differences (Shannon index) between the baseline and after 6 weeks were observed; however, fecal acetate levels were significantly increased. Another fundamental point in this study has been the developed understanding that the fecal microbiota composition at the baseline can be used to predict the responder or non-responder status. Metabolic responders are characterized by a higher abundance of *Subdoligranulum variabile* and *Dorea longicatena* in comparison with non-responders, whereas the abundances of *Eubacterium ventriosum* and *Ruminococcus torques* are lower in the baseline fecal samples of responders [100]. Moreover, Ng et al. (2021) studied microbiota engraftment after FMT in obese individuals with type 2 diabetes mellitus (T2DM). The authors enrolled 61 obese subjects with T2DM randomized and divided them into three parallel groups: FMT plus lifestyle intervention (LSI), FMT alone, and dummy transplantation plus LSI every 4 weeks for up to 12 weeks. The results showed that a group of subjects acquiring ≥20% of the lean microbiota profiles at week 24 were 100%, 88.2%, and 22% in the FMT plus LSI, FMT alone, and dummy plus LSI groups, respectively (*p* < 0.0001). Moreover, intriguing evidence indicated that combining LSI and FMT led to increases in *Bifidobacterium* and *Lactobacillus* compared with FMT alone (*p* < 0.05). The FMT plus LSI group exhibited reduced total and low-density lipoprotein cholesterol and liver stiffness at week 24 compared with the baseline (*p* < 0.05) [101]. The impact of FMT on the microbial population and the consequent improvements in clinical outcomes has also been evaluated in hepatic encephalopathy (HE). HE is a debilitating complication of cirrhosis, seriously affecting the quality of life of patients and caregivers [102]. HE has been associated with an increased relative abundance of ammonia-producing bacteria species. Subsequent hyperammonemia is associated with impaired neuronal function [103]. Specifically, in cirrhosis, marked reductions in beneficial commensal bacteria, such as Ruminococcaceae and Lachnospiraceae, and increases in non-autochthonous bacteria, such as Streptococcaceae and Enterobacteriaceae, have been observed [104]. Contemporary treatment strategies for HE consist of lactulose supplementation and treatment with antibiotics, such as rifaximin. Considering these aspects, FMT may represent a valid treatment option. The first case report of FMT in HE was published by Kao et al. in 2015. A fecal suspension delivered by an enema from an unrelated donor was infused into a recipient patient not able to tolerate lactulose treatment. The patient underwent four sessions on a weekly regimen. A marked improvement in clinical parameters was observed until week 14 when the patient reverted to their baseline condition [105]. These data suggest a beneficial but transient effect of FMT in HE. In 2017, Bajaj et al. published a paper in which patients with HE patients were treated with a single FMT via enema in addition to standard lactulose treatment [104]. In this trial, a single non-related donor was selected for their higher relative abundance of Lachnospiraceae and Ruminococcaceae. During a 5-month follow-up, fewer serious adverse events (2 versus 8) and new episodes of HE (0 versus 6) was observed in the FMT group (*n* = 10) compared with a control group (*n* = 10) solely receiving standard care. Furthermore, patients treated with FMT showed significant improvements in cognition and eubiosis. Specifically, after the FMT intervention, high levels of Lachnospiraceae and Ruminococcaceae were observed in the recipients.

Several aspects still need to be explored such as the optimal dose, administration route, and duration of benefits. The most crucial phase of FMT is the screening of donors. In fact, each facility must ensure safe infusion products in order to avoid any possible pathogen transmission and any clinical adverse events. For this purpose, as the guidelines suggest, the fecal matter should undergo strict screening by cultural, microscopy, and molecular analyses [75]. To date, the only case of a serious adverse effect has been the transmission of β-lactamase-producing *E. coli* (ESBL) in two patients receiving the same stool sample. Unfortunately, one of the patients died before the end of the program [4,106]. Future trials will answer these questions, furthering our understanding of how FMT impacts microbial populations and subsequent improvements in clinical outcomes. These findings substantiate the concept that fecal transplantation acts not only at the intestinal level, but can also orchestrate the complex metabolic network. Limitations highlighted by the studies presented here include the transient duration of the described beneficial effects. Therefore, evolutionary approaches to FMT, such as the use of capsules, represent a future direction in the treatment of intestinal and extra-intestinal disorders.

### 2.4. Engineered Bacteria for Shaping the Microbiota

The use of engineered bacterial strains represents the most futuristic and encouraging approach to modulating the structure and function of the gut microbiota. Such strains can be designed to interact with other commensal bacteria, influence metabolic pathways through the production of biomolecules, and modulate local and systemic immune functions [107,108]. These complex interactions have a strong impact on the host health/disease balance. The first step in this methodological strategy is selecting chassis bacteria on which to carry out the engineering [109]. The choice of the strain involves various aspects, such as the microenvironment, in which the bacterium will have to express its biological properties, bio-distribution and elimination kinetics, routes of administration, and biosafety criteria [108] (Figure 3).

Over the last decade, studies on engineered bacteria have boomed. Many of these studies focused on the treatment of intestinal and systemic disorders, in which alterations of the gut microbiota are a crucial factor in disease development. This section describes the main strains investigated in the clinical and pre-clinical phases and the diseases treated (Table 1).

The most chosen chassis is *Escherichia coli* Nissle 1917 (EcN). EcN 1917 has been considered a probiotic since its first isolation for the treatment of various gastrointestinal disorders. This bacterium can inhibit the growth of opportunistic pathogens, such as *Salmonella* spp. and other enterobacteria, through its interaction with the intestinal epithelium and the subsequent production of anti-inflammatory factors which contribute to the integrity of the intestinal barrier [110]. The major limitation of this strain as a probiotic is the short-lived intestinal colonization [111]. Researchers have tried to overcome this limitation by engineering the strain to achieve targeted effects despite the reduced colonization time. For example, Isabella and colleagues demonstrated the efficacy of an EcN 1917 strain in phenylketonuria (PKU) treatment in mice and primates [112]. PKU patients have a genetic deficiency of the enzyme complex involved in phenylalanine metabolism. The accumulation of this molecule leads to neurotoxicity [113]. The engineered EcN strain (SYNB1718) demonstrated an ability to degrade phenylalanine by lowering its levels [112]. A similar approach has been used in the treatment of another metabolic disorder related to hyperammonemia. Krutz et al. engineered a strain of EcN (SYNB1020) and fed it to mice and primates. They evidenced a significant drop in blood ammonium levels. The strain can increase arginine biosynthesis, thereby stimulating the consumption of ammonium. Further studies are currently ongoing to ascertain its biosafety in humans [114]. Another encouraging example of a metabolism intervention using an engineered EcN strain is the study by Chen and colleagues. In this case, a bacterium was engineered to produce *N*-acylphospatidylethanolamines, precursor molecules of anorexigenic fats. The study involved subjecting mice to a high-fat diet. The group of animals treated with the engineered EcN strain exhibited a marked reduction in fat mass compared with the group treated with the wild-type probiotic strain [115]. Diabetes is the most studied metabolic disease, on which much research has focused. Among the several desirable intervention strategies, the use of engineered strains is an extremely interesting topic. For example, in 2015, Duan et al. tested the impact of certain modified probiotics in diabetic mice. The bacterial strain tested was *Lactobacillus gasserii*. This strain was engineered to enable glucagon-like peptide (GLP-1) delivery to restore the insulin sensitivity of intestinal cells. The *L. gasserii* strain was modified to produce an inactive full-length version of the hormone capable of reprogramming intestinal cells into glucose-responsive insulin-secreting cells. Mice fed the engineered strain exhibited increased glucose tolerance and higher insulin levels than mice treated with the wild-type strain [116].

Another target in the use of this strategy is the inhibition of colonization by opportunistic pathogens. A complex network of interactions exists between commensal and opportunistic bacteria, namely, the “sense-kill system” [21]. Editing commensal strains to overexpress this property is an extremely interesting approach. In 2011, Saeidi evaluated the ability of engineered EcN 1917 to eradicate and prevent colonization by *Pseudomonas aeruginosa* in nematodes and mouse models [117]. Similar evidence was reported by Mao and colleagues. They demonstrated that an edited *Lactococcus lactis* strain detected the presence of molecules produced by *Vibrio cholerae* and increased the survival rate of infected animals [118].

It is now well established that the gut microbiota plays a key role in the modulation of the immune system and, thus, in local and systemic inflammation [119]. Influencing microbial populations and, consequently, immune networks, is an exciting challenge. The use of engineered bacteria is a related topic of great interest, motivating several studies. As early as 2006, for example, Braat et al. proposed a strain of *Lactococcus lactis* engineered to produce Il-10, an anti-inflammatory cytokine, in the treatment of diseases, such as Crohn’s disease [120]. The recent advances in molecular biology enabled several groups to use chassis bacteria not previously considered suitable for genetic manipulation. For example, CHAIN biotech [121] developed an engineered *Clostridium* strain capable of producing beta-hydroxybutyrate, an anti-inflammatory molecule. The use of spore-forming strains, such as *Clostridia,* overcomes the inherent limitations of probiotics, such as survival in acidic and gastric environments. Spores are resistant to stomach acid and enzymatic digestion in the upper GI tract. Once spores reach the terminal ileum and large intestine, they rapidly germinate and multiply [122]. The studies briefly described here clarify the advantages and limitations of live biotherapeutics. Designing and implementing engineered strains is the key challenge. It is necessary to consider factors such as the biological target, the site of activity, the desired effect duration, pharmacokinetic properties, patient biosafety, and manufacturing flexibility.

**Table 1 microorganisms-10-02424-t001:** A list of engineered bacteria used as potential therapeutic tools. The main target areas are metabolic pathways, the inhibition of colonization (sense–kill system), and immunity.

Biologic Target	Engineered Strain	Disorder	Reference
Metabolic pathway	*EcN SYNB1718* *EcN SYNB1020*	PKU Hyperammonemia	[112] [114]
	*EcN1917*	Obesity	[115]
	*L. gasserii*	Diabetes	[116]
Sense–Kill system	*EcN1917*	Opportunistic infections	[21]
	*Lactococcus lactis*	Opportunistic infections	[118]
Immune system	*L. lactis* *Clostridium strain*	IBD IBD	[120] [121]

## 3. Bacteriophages and Microbiota Shaping

Bacteriophages or, simply, phages, are natural predators of bacteria and represent the most ubiquitous and various biological groups on Earth, with an estimated population of 10^32^ [123]. Our body harbors billions of phages inside and on the surface which interact with bacterial populations and immune cells [124]. Several studies have demonstrated the key role of *bacterioma/phageoma* interactions in health/disease status [125,126]. As predators of bacteria, bacteriophages represent an intriguing therapeutic tool for “personalized medicine”. The history of phage therapy dates back to 1915, when the English bacteriologist Frederick Twort reported the presence of microorganisms capable of lysing bacterial cultures [127]. In 1917, the French microbiologist Felix d’Herelle understood that these were viruses, which he called “bacteriophages” [128]. With the advent of the antibiotic era (in the 1940s), attention to the therapeutic use of phages waned. This approach boomed after the Second World War, especially in Eastern Europe, where antibiotic recovery was tricky. The world’s leading center for the study and synthesis of bacteriophages is the George Eliava Institute in Tbilisi, Georgia [129]. In recent years, the efforts of researchers have focused on clarifying the interactions between bacterial and phage populations in the gut ecosystem. This relationship is extremely complex and can be mutualistic or competitive [125]; it is of great interest as a tool to manipulate the gut microbiota and achieve beneficial results for the host [130]. Several studies have attempted to demonstrate the therapeutic efficacy of phages. For example, Galtier and colleagues observed a marked reduction in adherent invasive *Escherichia coli* (AIEC) strains in patients with Crohn’s disease treated with phage therapy [131]. The eubiosis-restoring function of bacteriophages was described in another study, in which a phage-only filtrate was used as a replacement for classical FMT in CDI patients. The treatment proved effective in the five patients studied [132]. Other studies have focused on the role played by resident phage populations on the state of intestinal eubiosis. Specifically, the composition of intestinal viroma in patients with inflammatory diseases (IBDs) was explored. Crohn’s patients exhibit a completely different viroma from healthy patients. In order to understand the phage/bacteria relationship, Norman et al. assessed the relative abundance of *Faecalibacterium prausnitzii*, as reported above, a species known to be depleted in patients with IBD. In these patients, this bacterial depletion coincides with a marked increase in two natural predatory phage species of *F. prausnitzii* [133].

The precise modulation of microbiota via phage therapy was evaluated on another serious metabolic disease: alcoholic hepatitis. A recent study found that the proportions of *Enterococcus faecalis* cytolysin producers increased 2700-fold in the feces of patients with alcoholic hepatitis compared with healthy controls [134]. Patients who were positive for *E. faecalis* cytolysin producing showed a mortality of up to 89% compared with patients who were not colonized. Researchers used four phages selected from sewage to selectively kill the cytolytic *E. faecalis* and treat ethanol-induced liver injury and steatosis in germ-free mice. This evidence confirms data in the literature which presents the phageome and bacteriome as two dynamic and balanced entities [130]. In addition to the microbiome, the phageome composition is influenced by environmental factors and the host’s lifestyle habits, such as diet. There is much evidence to suggest that in subjects consuming a Western diet, the prevalence of two main phage taxa is observed, and their roles may be crucial in the balance of the gut microbiota, as seen below:Gokushovirinae are temperate phages exhibiting an ssDNA genome; andCrASS-like dsDNA phages can represent up to 90% of the phageome of a single individual.

Both taxa infect strains of *Bacteroides* spp. are key members of the healthy human microbiota and represent critical genera in the overall eubiotic setup of the host [125]. In this section, we present an overview of the impact of resident bacteriophages on the eubiotic status of the host and their potential use as a therapeutic tool. There are several advantages of phage therapy. Firstly, the specificity of intervention and the possibility of targeting specific bacterial species is potentially harmful to our microbiota. This specificity, coupled with the possibility of administering single doses of phages, makes this approach extremely attractive to replace or complement the use of antibiotics in some clinical situations [135,136,137]. Despite these encouraging aspects, some key points must be further investigated and clarified (Figure 4).

Deepening the complex interactions between bacteriophages, bacterial prey, and the immune system. Thousands of phage species have been identified but there is, currently, a dearth of public libraries and clinical trials. Further studies must be conducted to investigate the efficacy of the phage product and its biosafety for clinical use.

## 4. Clustered Regularly Interspaced Short Palindromic Repeats/Cas System (CRISPR/Cas)

Regarding gut microbiota manipulation, the possibility of employing a targeted approach using precise gene editing tools is remarkable. In recent years, several nuclease-based technologies have been developed to achieve targeted and efficient modifications at specific loci of interest in the bacterial genome. To date, the most promising system is to drive the Cas class of endonucleases via a short guide RNA (gRNA) to a specific target DNA sequence in order to modify it by inserting site-specific and unrepairable double-strand breaks (DBSs) [138]. The use of the CRISPR/Cas system could be a useful approach in clinical settings when considering the alarming increase in multidrug-resistant bacteria (MDR bacteria) in recent years. The use of broad-spectrum antibiotics mainly causes an imbalance in the gut microbiota populations. In this scenario, the CRISPR/Cas system may be used to selectively remove multidrug-resistant pathogens without disturbing commensal bacteria. In 2016, Kim et al. achieved the antibiotic re-sensitization of extended-spectrum beta-lactamase (ESBL)-producing *Escherichia coli* by transformation with an engineered plasmid containing both the sequence to encode Cas9 and the sequence encoding for the specific guide RNA to target the ampicillin resistance gene [139]. Although the specific target of the CRISPR/Cas system was the ampicillin gene, the bacteria were also found to be sensitive to cephalosporins, given that resistance genes are often contained within the same plasmid [139].

In another study by Rodrigues et al., it was found that constitutive expression of the pPD1 conjugative plasmid containing a CRISPR-Cas9-targeting cassette in *Enterococcus faecalis* enabled the selective removal of the *erm*B gene, which encodes erythromycin resistance, and the *tet*M gene, which encodes tetracycline resistance [140]. The authors reported that the use of these transformants significantly decreased the prevalence of erythromycin- and tetracycline-resistant *E. faecalis* in the intestine of the mouse model. Furthermore, *E. faecalis* was immune to the uptake of antibiotic resistance determinants, as if the CRISPR/Cas system provided additional protection from future acquisition [140].

In 2021, Neil et al. synthetized engineered probiotic strains capable of conjugation and CRISPR/Cas9 plasmid transmission [141]. This vehicle could eliminate strains of *Escherichia coli* resistant to several antibiotics (such as streptomycin, chloramphenicol, and tetracycline) from the mouse gut microbiota with a single dose of treatment [141].

In a recent study by He et al., the use of CRISPR/Cas9 was combined with a transposon system to specifically delete both the plasmid mcr-1 colistin resistance gene and chromosomal mcr-1 in *Escherichia coli*, leading to the reversion of antibiotic susceptibility and the development of immunity against the acquisition of the exogenous mcr-1-containing plasmids in bacteria that were already susceptible [142].

Another target of the CRISPR/Cas system on microbial gene editing might be the deletion of virulence genes. In 2014, Bikard et al. showed that transfection of the CRISPR/Cas system by bacteriophages into a population of *Staphylococcus aureus* enabled the selective deletion of virulence genes, leading to the death of expressing strains. In addition, this method has been used to target antibiotic resistance genes, resulting in the elimination of staphylococcal resistance plasmid genes and the immunization of non-virulent *Staphylococcus* spp., preventing the horizontal transfer of the plasmid resistance gene. This approach showed efficacy in an in vivo mouse model with staphylococcal skin colonization [143]. Another experiment concerning virulence gene targeting was carried out by Citorik et al., who described significant decolonization of the enterohemorrhagic pathogenic strain of *Escherichia coli* (EHEC) in *Galleria mellonella* larvae by deletion of the virulence gene *eae*, encoding for intimin, the major virulence determinant of this pathogenic strain [144].

Given these findings, the CRISPR/Cas9 system appears to be a promising tool for the targeted in situ manipulation of the microbiome. However, more studies are needed in order to evaluate the efficiency and safety of human body cells in vivo.

## 5. Conclusions

The gut microbiota appears to be a key factor in an individual’s health/disease balance. Delineating its composition represents the first step toward personalized diagnostic approaches. Once the extent of the disorder is understood, shaping, modifying, or, in some cases, silencing an individual’s gut microbial community lays the foundation for personalized therapies. To this end, FMT by colonoscopy or capsule administration represents the approaches that, to date, have been most widely studied and clinically evaluated on human models. Moreover, in recent years, fecal microbiota infusion has become an increasingly standardized method, enabling many research groups to organize clinical trials. A fundamental aspect in prospective research might be the use of all these strategies as “combined therapies”, in order to take advantage of the various beneficial features of each of them.

## Figures and Tables

**Figure 1 microorganisms-10-02424-f001:**
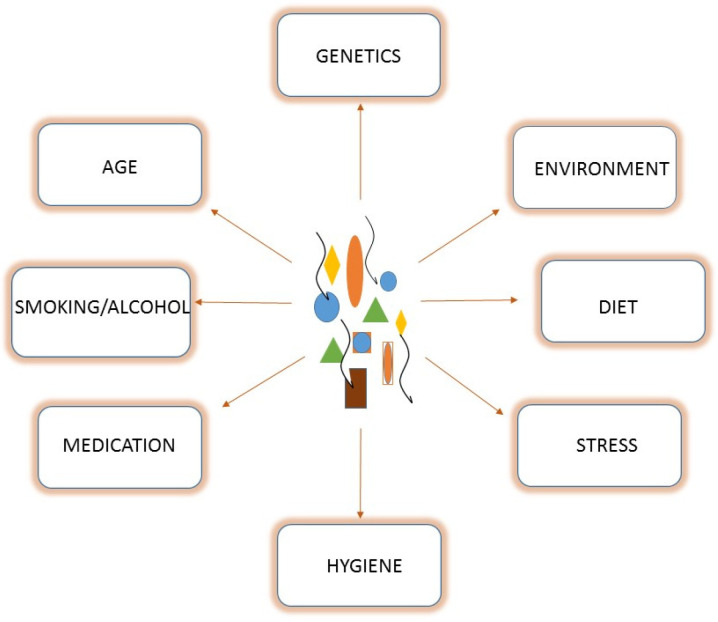
Microbiota diversity and richness are strongly influenced by the host and environmental factors.

**Figure 2 microorganisms-10-02424-f002:**
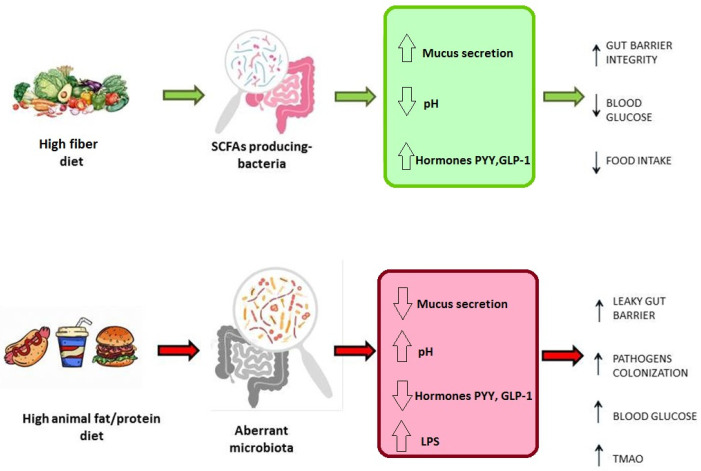
Impact of diet on intestinal microbiota and health outcomes. SCFA = short chain fatty acids; PYY = peptide YY; GLP-1 = glucagon-like peptide 1; LPS = lipopolysaccharide; TMAO = trimethylamine *N*-oxide.

**Figure 3 microorganisms-10-02424-f003:**
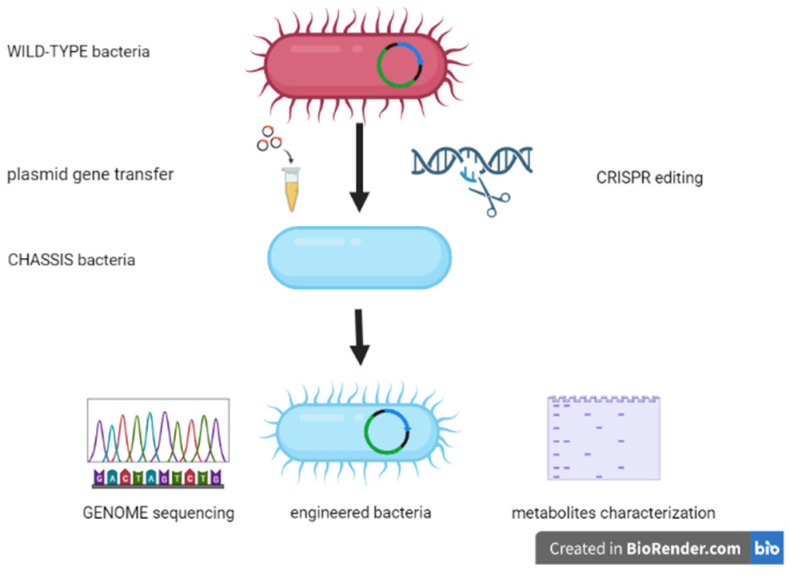
Engineering steps for a candidate chassis wild-type bacterium.

**Figure 4 microorganisms-10-02424-f004:**
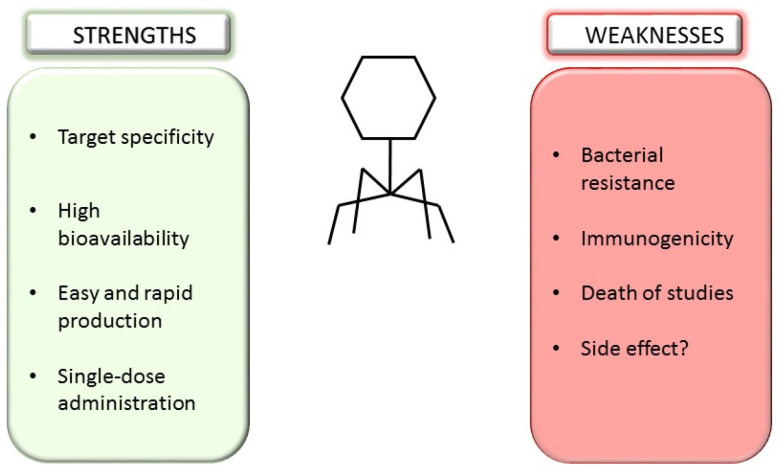
Strengths and weaknesses of phage therapy.

## Data Availability

Not applicable.

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
