# Peer review of "Fecal Microbiota Transplantation and Other Gut Microbiota Manipulation Strategies"

_microorganisms, 2022, doi:10.3390/microorganisms10122424_

Round 1
Reviewer 1 Report
Dear Editor,
I have read with great interest the manuscript submitted by Quaranta et al. entitled “Fecal Microbiota Transplantation and other gut microbiota manipulation strategies”. There are some issues that need to be addressed before further processing.
As a general concept, I suggest revision by a native English speaker.
Section: 1. Introduction. I suggest two recent papers to include that covers general aspect on gut microbiota given the target of this review paper (DOI1: https://doi.org/10.1152/ajpgi.00161.2019 // DOI2: 10.3390/jcm9113705 ).
Section: 2. Microbiota-changing interventations (the term “interventation” is not English btw).
Lines 46-54: the authors should add some information regarding the age when the microbiota becomes “stable”, introduce the concept of eubiosis/dysbiosis and report the most prevalent bacterial phyla.
2.1 Diet: the authors report only the concept of SCFAs, however they should add some hints about other important things that can be modulated by diet (bile acids, etc). Also, add the controversy on SCFAs and tumorigenesis.
2.2 Probiotics, prebiotics, synbiotics, postbiotics. The authors report only the pros of probiotics. They should also report data about the efficacy (i.e., supplemented strains start to be less prevalent just after therapy, possible yeast contamination, etc) – check DOI1. In the prebiotics section, I would add more information on how they can potentiate SCFAs production and action.
2.3 The authors should also report data on death after FMT (check DOI1
Author Response
Dear Editor and Reviewers,
Thanks for your suggestions for the improvement of our manuscript. Following you can find (traced in red) all corrections, additions and stylistic changes.
Response to Reviewer 1
As a general concept, I suggest revision by a native English speaker
We have provided extensive english editing by MDPI English Editing service ID: English-55150
Section: 1. Introduction. I suggest two recent papers to include that covers general aspect on gut microbiota given the target of this review paper (DOI1: https://doi.org/10.1152/ajpgi.00161.2019 // DOI2: 10.3390/jcm9113705 ).
We have expanded the introduction section with detailed description on gut microbiota composition, using your suggested papers.
Section: 2. Microbiota-changing interventations (the term “interventation” is not English btw).
Done
Lines 46-54: the authors should add some information regarding the age when the microbiota becomes “stable”, introduce the concept of eubiosis/dysbiosis and report the most prevalent bacterial phyla.
We have added suggested concepts in lines 99-105
2.1 Diet: the authors report only the concept of SCFAs, however they should add some hints about other important things that can be modulated by diet (bile acids, etc). Also, add the controversy on SCFAs and tumorigenesis.
We have added suggested concepts in lines 156-161 and 172-191 and 289-303.
2.2 Probiotics, prebiotics, synbiotics, postbiotics. The authors report only the pros of probiotics. They should also report data about the efficacy (i.e., supplemented strains start to be less prevalent just after therapy, possible yeast contamination, etc) – check DOI1. In the prebiotics section, I would add more information on how they can potentiate SCFAs production and action.
We have added suggested concepts in lines 261-272.
2.3 The authors should also report data on death after FMT (check DOI1).
Done in lines 525-531
Reviewer 2 Report
Review: Fecal Microbiota Transplantation and other gut microbiota ma-2 nipulation strategies.
Title: Please add the type of review in the title.
General comment: Please review the publication guidelines and correctly place references throughout the text.
Abstract: The authors should point out the most relevant results of this review.
General comment: Authors should review citations throughout the text, and cite authors as they appear in the text and not at the end of the explanation of the study.
Introduction:
- Please organize the text into paragraphs to make the reading flow more smoothly.
- In what way have they been related? : “Several studies have shown bacterial unbalance associated to various in-30 testinal and systemic disease such as obesity, inflammatory bowel diseases (IBD), cardio-31 vascular, neurological and psychiatric disorders [5]”
- The introduction is not very fluid and mixes ideas. Perhaps the authors can put a paragraph with the presentation of the subject and the microorganisms. Another paragraph with previous evidence on the topic, which should be developed further. And finally the justification and relevance of this work, ending with the objective of the study.
- Add reference: “In fact, in case of a vaginal delivery, early colonizers originate from the 48 mother’s vaginal and fecal microbiota such as Lactobacillus, Prevotella, Bacteroides, Esch-49 erichia/Shigella”
- Add reference: These factors highlight an intrinsic dynamism of the gut microbiota expressed 58 through a gradual process. The synergistic action of these elements leads to a metabolic 59 state of health or disease..
- that has been described before? Reference “ changing intervention can be 62 classified in two metodological strategies”
- Add reference: “Crucial elements of this strategy are diet, administration of prebiotics, probiotics and fecal microbi-68 ota transplantation (FMT).”:
- What trials? Add reference: Moreover, some trials were 73 focused on the use of bacteria for the drugs delivery and on phage therapy in order to 74 delete specific “bad” bacteria population.
- Add reference: A short-term diet can induce a shift in the gut population, 121 but these changes appear to be transients.
- Please, explain this. Their use as a therapeutic tool includes numerous advantages 217 but also strong limitations.
- Please add reference to support this statement.
- Please add reference An interesting study has 257 been conducted by Kim et al in 2022. The scien-250 tific literature has produced numerous papers demonstrating a close link between gut 251 bacteria and the development and progression of circulatory diseases such as atheroscle-252 rosis and hypertension.
- Add reference: Zhong et al 272 in 2021 conducted a study demonstrating that microbiota transplantation from healthy 273 donors can decrease blood pressure in patients with Hypertension
- the authors point to several, but only reference one study. Add or modify sentence. “Several evidences have suggested that gut microbiota composition 283 play a fundamental role in gastrointestinal and in neurodevelopmental dysfunction in 284 Autism Spectrum Disorder (ASD) patients [69]”.
- Add reference Kim et al. achieved antibiotic re-sensitization of extended-spectrum beta…-
- Add reference “In another study by Rodrigues et al. it was seen that constitutive…”
- Add reference In 2021, Neil et al. synthetized engineered probiotic strains capable of conjugation 588 and CRISPR/Cas9 plasmid transmission.
- Add reference In a recent study by He et al., the use of CRISPR
Author Response
Dear Editor and Reviewers,
Thanks for your suggestions for the improvement of our manuscript. Following you can find (traced in red) all corrections, additions and stylistic changes.
Title: Please add the type of review in the title.
We did not found informations about title review to use, exept the form "Review"
General comment: Please review the publication guidelines and correctly place references throughout the text.
Done
Abstract: The authors should point out the most relevant results of this review.
Done in lines 18-24
General comment: Authors should review citations throughout the text, and cite authors as they appear in the text and not at the end of the explanation of the study.
Done
- Introduction: Please organize the text into paragraphs to make the reading flow more smoothly.
We have provided two subparagraph: 1.1 and 1.2
- In what way have they been related? : “Several studies have shown bacterial unbalance associated to various intestinal and systemic disease such as obesity, inflammatory bowel diseases (IBD), cardio-31 vascular, neurological and psychiatric disorders [5]”
We have modified the sentence in line 356-359
- The introduction is not very fluid and mixes ideas. Perhaps the authors can put a paragraph with the presentation of the subject and the microorganisms. Another paragraph with previous evidence on the topic, which should be developed further. And finally the justification and relevance of this work, ending with the objective of the study.
We have expanded the introduction section with detailed description on gut microbiota composition, using your suggested papers. Moreover introduction section has been divided in two subsections in order to make this part more fluid. Il lines 85-88 we justify the potential relevance of our paper.
- Add reference: “In fact, in case of a vaginal delivery, early colonizers originate from the mother’s vaginal and fecal microbiota such as Lactobacillus, Prevotella, Bacteroides, Escherichia/Shigella”
Done
- Add reference: These factors highlight an intrinsic dynamism of the gut microbiota expressed through a gradual process. The synergistic action of these elements leads to a metabolic state of health or disease.
Done
- That has been described before? Reference “ changing intervention can be classified in two metodological strategies”
Done, Ref 20: Fan, Y.; Pedersen, O. Gut microbiota in human metabolic health and disease. Nat Rev Microbiol. 2021;19(1):55-71. doi:10.1038/s41579-020-0433-9
- Add reference: “Crucial elements of this strategy are diet, administration of prebiotics, probiotics and fecal microbiota transplantation (FMT).”
Done
- What trials? Add reference: Moreover, some trials were focused on the use of bacteria for the drugs delivery and on phage therapy in order to delete specific “bad” bacteria population.
We have added reference 21: Lee, H.L.; Shen, H.; Hwang, I.Y.; Ling, H.; Yew, W.S.; Lee, Y.S.; Chang MW. Targeted Approaches for In Situ Gut Microbiome Manipulation. Genes (Basel). 2018;9(7):351. doi: 10.3390/genes9070351.
- Add reference: A short-term diet can induce a shift in the gut population, but these changes appear to be transients.
Done
Please, explain this. Their use as a therapeutic tool includes numerous advantages but also strong limitations.
Done in lines 261-272
- Please add reference to support this statement.
Done with Ref. 58-59-60
- Please add reference An interesting study has been conducted by Kim et al in 2022. The scientific literature has produced numerous papers demonstrating a close link between gut bacteria and the development and progression of circulatory diseases such as atherosclerosis and hypertension.
Done
- Add reference: Zhong et al 272 in 2021 conducted a study demonstrating that microbiota transplantation from healthy 273 donors can decrease blood pressure in patients with Hypertension
Done
- The authors point to several, but only reference one study. Add or modify sentence. “Several evidences have suggested that gut microbiota composition play a fundamental role in gastrointestinal and in neurodevelopmental dysfunction in Autism Spectrum Disorder (ASD) patients [69]”.
We added also two references: 91 and 92
- Add reference Kim et al. achieved antibiotic re-sensitization of extended-spectrum beta…
Done
- Add reference “In another study by Rodrigues et al. it was seen that constitutive…”
Done
- Add reference In 2021, Neil et al. synthetized engineered probiotic strains capable of conjugation 588 and CRISPR/Cas9 plasmid transmission.
Done
- Add reference In a recent study by He et al., the use of CRISPR
Done
Rome 2/12/2022
Alessandra Guarnaccia
Round 2
Reviewer 1 Report
The authors have edited the manuscript according to the reviewers' suggestions. The manuscript is not acceptable for publication.